# Learning to Grasp the Ungraspable with Emergent Extrinsic Dexterity

**Wenxuan Zhou**
Robotics Institute
Carnegie Mellon University
wenxuanz@andrew.cmu.edu

**David Held**
Robotics Institute
Carnegie Mellon University
dheld@andrew.cmu.edu

**Abstract:** A *simple* gripper can solve more *complex* manipulation tasks if it can utilize the external environment such as pushing the object against the table or a vertical wall, known as "Extrinsic Dexterity." Previous work in extrinsic dexterity usually has careful assumptions about contacts which impose restrictions on robot design, robot motions, and the variations of the physical parameters. In this work, we develop a system based on reinforcement learning (RL) to address these limitations. We study the task of "Occluded Grasping" which aims to grasp the object in configurations that are initially occluded; the robot needs to move the object into a configuration from which these grasps can be achieved. We present a system with model-free RL that successfully achieves this task using a simple gripper with extrinsic dexterity. The policy learns emergent behaviors of pushing the object against the wall to rotate and then grasp it without additional reward terms on extrinsic dexterity. We discuss important components of the system including the design of the RL problem, multi-grasp training and selection, and policy generalization with automatic curriculum. Most importantly, the policy trained in simulation is zero-shot transferred to a physical robot. It demonstrates dynamic and contact-rich motions with a simple gripper that generalizes across objects with various size, density, surface friction, and shape with a 78% success rate. Videos can be found at https://sites.google.com/view/grasp-ungraspable.

**Keywords:** Manipulation, Reinforcement Learning, Extrinsic Dexterity

## 1 Introduction

Humans have dexterous multi-fingered hands; however, similarly dexterous robot hands are expensive and fragile. Instead, a simple hand can achieve dexterous manipulation by leveraging the environment, known as "Extrinsic Dexterity" [1]. For example, a simple gripper can rotate an object in-hand by pushing it against the table [2], or lifting an object by sliding it along a vertical surface [3]. With the exploitation of external resources such as contact surfaces or gravity, even simple grippers can perform skillful maneuvers that are typically studied with a multi-fingered dexterous hand. Different from a common practice of considering the robot and an object of interest in isolation, extrinsic dexterity focuses on a holistic view of considering the interactions among the robot, the object, and the external environment.

Previous work in extrinsic dexterity has demonstrated a variety of tasks such as in-hand reorientation with a simple gripper, prehensile pushing, or shared grasping [1, 2, 3]. However, the underlying approaches come with several limitations such as relying on hand-designed motions or primitives with limited capability of generalizing across different objects, making assumptions about contact locations and contact modes, or requiring specific gripper design [1, 2, 3, 4, 5, 6, 7]. Instead, we build a system with reinforcement learning (RL) to remove these limitations. With RL, the agent can learn a closed-loop policy of how the robot should interact with the object and the environment to solve the task, taking into account both planning and control. In addition, when trained with domain randomization, the policy can learn to be robust to different variations of physics. These properties of RL can enable extrinsic dexterity in a more general setting.

In this work, we study "Occluded Grasping" as an example of a task that requires extrinsic dexterity. The goal of this task is to grasp an object in poses that are initially occluded. Consider, for example, a robot that needs to grasp a cereal box lying on its side on a table; the desired grasp is not reachable

6th Conference on Robot Learning (CoRL 2022), Auckland, New Zealand.

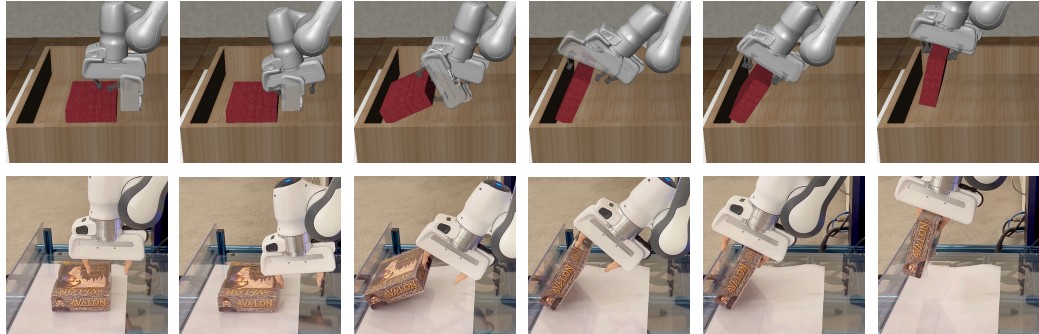

Figure 1: We present a system for the "Occluded Grasping" task with extrinsic dexterity. The goal of this task is to reach an occluded grasp configuration (indicated by a transparent gripper). The figure shows an example of the emergent behavior of the policy and successful sim2real transfer.

because it is partially occluded by the table (Figure 1). To achieve this grasp with a parallel gripper, the robot might rotate the object by pushing it against a vertical wall to expose the desired grasp and then reach it. This task is in contrast with common grasping tasks which focus on reaching an unoccluded grasp in free space with a static or near-static scene [8, 9, 10].

The goal of this work is to build a system for the "Occluded Grasping" task as an example of the combination of RL and extrinsic dexterity that works on a physical robot. We investigate design choices of such a system and emphasize the simplicity of the method. With model-free RL, we design a reward function that optimizes pre-grasp and grasping motion without the separation of stages as previous work in pre-grasp [11, 12, 13]. By placing the object in the bin and using a compliant low-level controller, the agent shows emergent extrinsic dexterity behavior without additional reward terms. We also incorporate a set of desired grasps with a training curriculum and a grasp selection procedure during evaluation. We improve the policy with Automatic Domain Randomization [14] over physical parameters which robustify the contact-rich behaviors across noise and environment variations. In the experiments, we provide a comprehensive evaluation of the system in simulation to analyze the importance of each component. The policy is zero-shot transferred to the physical robot and successfully executes similar behaviors to complete the task. The policy achieves a success rate of 78% and shows generalization across various out-of-distribution objects. Our main contribution is the real robot experiment results. Existing work with a simple hand has not shown such behaviors on the real robot with a similar level of complexity in contact events and generalization across objects at the same time.

## 2 Related Work

### 2.1 Extrinsic dexterity

"Extrinsic dexterity" is a type of manipulation skill that enhances the intrinsic capability of a robot hand using external resources including external contacts, gravity, or dynamic motions of the arm [1]. Previous work in extrinsic dexterity has demonstrated complex tasks with a simple gripper including in-hand reorientation [1, 4], prehensile pushing [2, 5], shared grasping [3], and more. Their methods are based on hand-crafted trajectories [1], task-specific motion primitives [3, 4, 15], or planning over contact modes [2, 5, 6, 7] to simplify the problem. They relies on careful assumptions on contacts such as assuming a fixed number of sticking contacts between the fingertips and the object. In this work, we take an alternative approach to use RL to learn a closed-loop policy that considers both planning and control without limitations on contact events. The resulting policy shows more versatile contact switches beyond prior work and can be transferred to a physical robot.

### 2.2 Grasping

Grasping is an important task in robot manipulation and has been studied from various aspects.

**Grasp generation:** One area of study in grasping is to generate stable grasps using analytical approaches [16, 17] or learning approaches [8, 9, 18, 19, 20]. In our paper, we assume that the desired grasp configurations are given which may come from any grasp generation method.

**Grasp execution:** To execute a grasp following grasp generation, a motion planner is usually used to generate a collision-free path towards the desired grasp configuration [10, 21, 22, 23]. All of these works aim at achieving the unoccluded grasp configurations in static or near-static scenes. Instead,

our work focuses on a complementary direction of achieving occluded grasp locations by interacting with the object of interest. Another line of work in grasping uses an end-to-end pipeline without the separation of grasp generation and grasp execution [24, 25]. However, they do not demonstrate performing the occluded grasps studied in this work.

**Pre-Grasp manipulation:** To deal with occluded grasps, prior work has studied pre-grasps as a preparatory stage of the grasping task. Typical motions for pre-grasps include rotation through planar pushing [12], sliding the object to the edge of the table [13, 26], or rotate the object against the wall [11]. Sun et al. [11] is the most related to our work, but they use a specially designed end-effector to perform the pre-grasp motion and then use a second gripper to grasp. We demonstrate that the full grasping task can be solved with a single gripper without special requirements on the end-effectors. These previous work typically separates pre-grasp motion and grasp execution into two stages and impose restrictions on the transitions of the stages. Instead, we co-optimize pre-grasp and grasp execution throughout the episode without explicit separation of the stages.

### 2.3 Reinforcement Learning for Manipulation

Previous work in RL for manipulation usually treats the object and the robot in isolation from the environment without considering extrinsic dexterity. RL has been applied to dexterous manipulation with a multi-fingered hand and shows contact-rich behaviors [14, 27, 28]. In contrast, with a parallel gripper, prior work focuses on tasks with limited contacts and object motions without utilizing the environment [29, 30]. This is the first work that demonstrates extrinsic dexterity with a simple parallel gripper using RL.

## 3 Task Definition: Occluded Grasping

The goal of a common grasp execution task is to move the end-effector $E$ close to a given desired grasp pose $g$. The desired grasp might come from any grasp generation method [8, 9, 20] as the input to the grasp execution task. As shown in Figure 2, we define an **"Occluded Grasping"** task to be a subset of the grasp execution tasks where the input desired grasp $g$ is initially occluded. To clarify, the term "occluded" in this work is more than visual occlusion. It means the desired grasp intersects with the table and moving the gripper in free space cannot solve this task. The robot has to interact

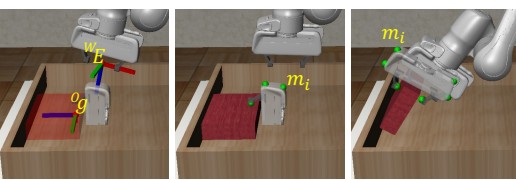

Figure 2: Notations: $^W E$ denotes the pose of the end-effector in the world frame $^W$. $^O g$ denotes the target grasp in the object frame $^O$. Six marker locations $m_i$ in green on the target grasp are used to calculate the occlusion penalty.

with the object to make the grasp pose reachable. The grasp $^O g$ is defined in the object frame $O$ and moves with the object. Formally, the grasp execution is defined to be successful if the position difference $\Delta T(g, E)$ and the orientation difference $\Delta \theta(g, E)$ are less than the pre-defined thresholds $\epsilon_T$ and $\epsilon_\theta$ respectively at the end of an episode. After successfully reaching a desired grasp pose, the gripper will be closed to complete the grasp. In addition, when the input to the system is a set of grasps $G = \{g_i\}_{i=1}^k$ instead of a single grasp, the agent may select any of the grasp to approach to.

## 4 Learning dexterous grasping with Reinforcement Learning

We build a system that learns a closed-loop policy for the occluded grasping task defined above with model-free RL. In this section, we will discuss important design choices of the system including the design of the RL problem, the extrinsic environment, and the choice of low-level controller. Then we will discuss how to deal with a set of grasps by training with a grasp curriculum and selecting the best grasp during evaluation. We also include Automatic Domain Randomization [14] to improving the generalization of the policy across environment variations.

### 4.1 Preliminaries: Goal-conditioned Reinforcement Learning

We define a Markov Decision Process (MDP) with states $s_t \in \mathcal{S}$, actions $a_t \in \mathcal{A}$, reward function $r : \mathcal{S} \times \mathcal{A} \to \mathbb{R}$, and discount factor $\gamma$. The state space, action space and the reward function for our task will be discussed in detail in the next section. The goal is to find a policy $\pi(a_t|s_t)$ that maximizes the return $R_t = \sum_{k=t}^{\infty} \gamma^{k-t} r(s_k, a_k)$. A Q-function is defined to be the expected return

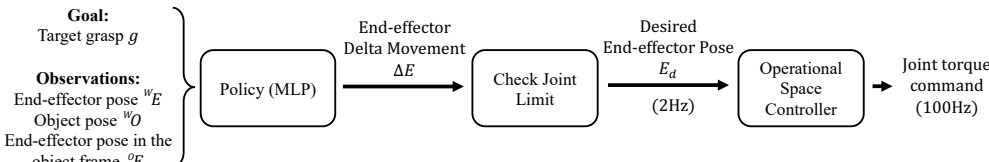

Figure 3: Outline of policy execution: Given the observations, the policy outputs an end-effector delta movement (Section 4.2) to the low-level controller (Section 4.4).

of the policy $Q^\pi(s, a) = \mathbb{E}_\pi[R_t | s_t, a_t]$. In goal-conditioned RL, we define a set of goals $\eta \in \mathcal{G}$ correspond to the reward function $r_\eta(s_t, a_t)$ [31]. To train a policy with a set of goals, both the policy and the Q-function will now take the goal $\eta$ as input, given by $\pi(a_t | s_t, \eta)$ and $Q^\pi(s_t, a_t, \eta)$. In the occluded grasping task, we use the desired grasps as goals.

## 4.2 RL Problem Design

**Observation and Action Space:** The observation that is input to the policy includes a target grasp configuration in the object frame $^O g$, the pose of the end-effector in the world frame $^W E$ and the object pose in the world frame $^W O$. Note that the policy only takes one grasp $^O g$ as input but we will discuss how to deal with a set of grasps in Section 4.5. For real robot experiments, we use Iterative Closest Point (ICP) for pose estimation of the object which matches a template point cloud of the object to the current point cloud [32]. The action space of the policy is the delta pose of the end-effector $\Delta E$ in its local frame which is passed into a low-level controller (Section 4.4). An outline of the policy execution is shown in Figure 3. More details can be found in Appendix B.

**Reward:** We design the reward function to optimize the pre-grasp motion and grasp execution without separating them into two stages as in previous work [11, 13, 12, 26]:

$$r = \alpha D(g, E) + \beta \sum_i P(m_i) \quad \text{(1a)} \qquad D(g, E) = \alpha_1 \Delta T(g, E) + \alpha_2 \Delta \theta(g, E) \quad \text{(1b)}$$

where $\alpha_1$, $\alpha_2$ and $\beta$ are the weights for the reward terms. The first term of Equation 1a, $D(g, {}^O E)$, is the pose difference between the target grasp and the current end-effector pose, which is to optimize for reaching the grasp. This term is expanded in Equation 1b to include the translational and rotational distance, as described in Section 3. The second term of Equation 1a is the target grasp occlusion penalty which is to penalize the agent if the target grasp configuration is in collision with the table. This corresponds to a pre-grasp objective. To measure how much the target grasp is occluded by the table, we set six marker points on the target gripper (Figure 2) denoted as $m_i$ and compare the height of the markers with the table top. If a marker is below the table top, the height difference will be used as the penalty. Including this occlusion penalty can effectively reduce the local optima where the gripper will reach close to the target grasp (which is occluded) without trying to move the object. Note that we did not impose any reward terms that are explicitly related to extrinsic dexterity. In our system, the use of extrinsic dexterity is an emergent behavior of policy optimization given our objective and environmental setup.

## 4.3 Extrinsic Environment

To exploit the benefits of extrinsic dexterity from object-scene interaction in this task, we construct the scene as having an object in a bin, instead of leaving the object on the table as shown in Figure 2. We also make the workspace of the robot large enough such that the robot can move the object to make contacts with the walls (during which the robot itself may also make contact with the wall). In the experiments, we will show that the policy will learn to utilize the wall to rotate the object. Without the wall, it is not able to find a strategy that can perform the task with the parallel gripper.

## 4.4 Choice of Low-level Controller

The choice of low-level controller is important for this task due to the fact that we expect the agent to use extrinsic dexterity which involves rich contacts among the gripper, the object and the bin. We choose Operation Space Control (OSC) as the lower-level controller to execute the policy output which operates at a higher frequency (100Hz) than the RL policy (2Hz) [33] (Figure 3). Given a desired pose of the end-effector, OSC first calculates the corresponding force and torque at the end-effector to minimize the pose error according to a PD controller with gain $K_p$ and $K_d$. Then, the desired force and torque of the end-effector will be converted into desired joint torques according to

the model of the robot. We choose relatively low gains so that the controller becomes compliant in the end-effector space. There are two benefits of a compliant OSC in such a contact-rich manipulation task with extrinsic dexterity. First, being compliant in end-effector space allows safe execution of the motions without smashing the gripper on the objects or the bin. Limiting the delta pose and selecting proper gains $K_p$, $K_d$ will limit the final force and torque output of the end-effector. If we use a controller that is compliant in the joint configuration space instead, we will not have direct control over the maximum force the end-effector might have on the object and the bin. Second, as shown in Martín-Martín et al. [34], using OSC as the low-level controller might speed up RL training and improve sim2real transfer for contact-rich manipulation.

## 4.5 Multi-grasp Training and Grasp Selection

In this section, we consider the scenario in which a set of desired grasp configurations are given instead of just a single one. During training, given a set of grasps $G_{train}$, we aim at covering as many grasp configurations as possible. As we will show in the experiments, reaching different grasps might require a significantly different behavior. Learning directly over a diverse set of goals might create difficulties for policy learning [35, 36]. We use an automatic curriculum following OpenAI et al. [14] to gradually expand the set of grasps to be trained with. We start the training with just a single fixed grasp; after the policy has achieved a success rate larger than a threshold, it will be trained on a slightly larger set with grasps close to the initial grasp location.

During testing, if a set of grasps $G_{test}$ is provided, we can select the best grasp within the set to improve the performance of the grasping task, following previous work in integrated grasp and motion planning [10, 21, 22]. With value-based model-free RL algorithms such as Soft Actor Critic [37], the policy is trained together with a Q-function (defined in Section 4.1). We propose to select the grasp that maximizes the learned Q-function for the given observation and action: $g^* = \arg\max_{g \sim G_{test}} Q(s_t, a_t, g)$. The selection can be performed for each timestep $t$, or at the beginning of an episode when $t = t_0$. We include both implementations in the experiments. This can select the grasp that is most easily reached which depends both on the environmental configuration as well as how well the policy has learned to achieve different grasp configurations.

## 4.6 Improving Policy Generalization

To improve generalization across environment variations, we train the policy with Automatic Domain Randomization (ADR) [14]. Similar to the multi-grasp curriculum, the policy is first trained on a single environment with a single object, and gradually expands the range of randomization automatically according to its performance. This significantly reduces the effort of tuning the range of domain randomization. We randomize over different variations of the environment properties such as object size, density, and friction coefficient. We also randomize the parameters of the controller to improve sim2real transfer. More descriptions on the ADR procedure can be found in Appendix C.

## 5 Experiments

We build the simulation environment for this task using Robosuite [38] and the MuJoCo simulator [39] as shown in Figures 1 and 2. The environment contains a Franka Emika Panda robot with a parallel gripper and an object in the bin in front of it. We focus on grasping large flat objects from the side since they cannot be grasped with a top-down motion. The policy is trained in simulation with Soft Actor Critic [37]. In this section, we include the results in simulation to discuss each component of the proposed system. We then evaluate zero-shot sim2real transfer on a physical Panda robot across different objects. Implementation details can be found in Appendix B.

### 5.1 Training Curves and Ablations

We first evaluate our method by training the policies with a single desired grasp in the default environment without ADR. Figure 4a shows the training curve of the proposed method and the ablations. The policy trained with the complete system can reach a success rate of $100\%$ before 4000 episodes which corresponds to 160000 environment steps. To evalu-

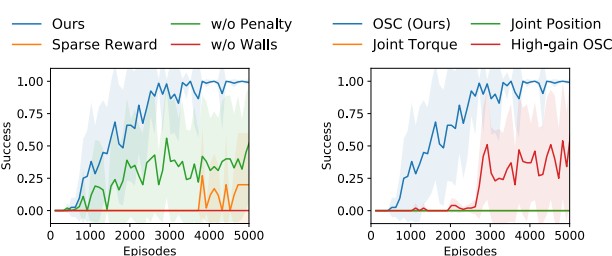

Figure 4: Training curves and ablations: **(a)** ablations on the reward function and the wall. **(b)** ablations on the controller.

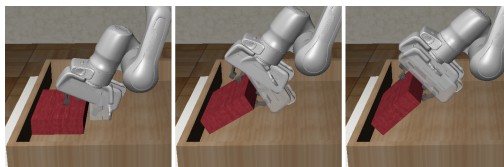

(a) **Local optima:** An example of local optima where the gripper uses the bottom finger to lift the object (instead of the top) and then fails to move the object between its two fingers to prepare for the grasp.

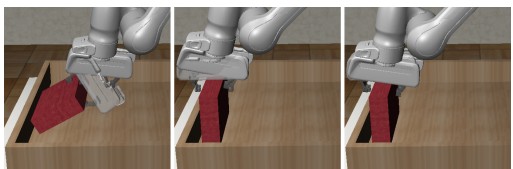

(b) **Standing object:** One of the successful strategies is to flip the object until it stands on the side and then reach the grasp.

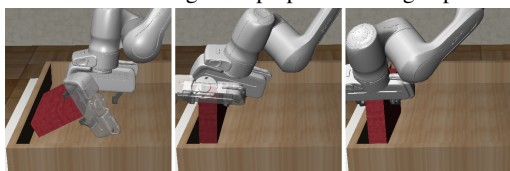

(c) **MultiGrasp-Front:** When the desired grasp is at the corner, the policy flips the object by pushing it on the side and then move close to the grasp.

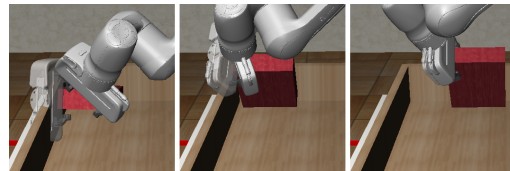

(d) **MultiGrasp-Side:** When the grasp is on the side, the policy can use another side of the wall to rotate the object and reach the desired grasp.

Figure 5: Visualizations of the policies in different scenarios.

ate the importance of extrinsic dexterity, we remove the walls of the bin. The resulting policies have $0\%$ success rate and push the object outside of the table. For ablations on the reward function, we remove the occlusion penalty (the second term of Equation 1a) and also try a sparse reward. Without the occlusion penalty, the policy is more likely to get stuck at a local optima (an example shown in Figure 5a) and thus the success rate becomes lower. With the alternative of a $\{-1, 0\}$ sparse reward, the policy learns much slower. We also experiment with different low-level controllers. Both joint torque and joint position control lead to worse performance which indicates the importance of using end-effector coordinates. With a less compliant controller by increasing the gains of OSC, the success rate becomes lower which demonstrates the importance of compliance for contact-rich tasks in addition to the safety considerations.

## 5.2 Emergent Behaviors

Figure 1 shows a typical strategy of a successful policy which involves multiple stages of contact switches. The gripper first moves close to the object and makes contact on the side of the object with the top finger. It then pushes the object against the wall to rotate it. During this stage, the gripper usually maintains a fixed or rolling contact with the object, but sliding also occurs. The object might have sliding or sticking contacts with the wall and the ground. After the gripper has rotated a bit further and the bottom fingertip is below the object, the gripper will let the object drop on the bottom finger. After that, the gripper will try to match the desired pose more precisely. At this point, the policy has executed the grasp successfully and it is ready to close the gripper. This type of learned contact-rich behaviors with a simple gripper has not been shown in previous work. In Section 5.5, we will further demonstrate that it can be transferred to a physical robot.

One of the key decisions in this strategy is to use the top finger to rotate the object instead of the bottom finger. One might suppose an alternative approach which is to use the bottom finger to scoop the object against the wall and then directly roll the finger underneath the object to reach the grasp. However, this strategy is not physically feasible on the parallel gripper due to the limited degree of freedom of the finger. We observe that the policies that follow this strategy during exploration usually get stuck at a local optima without successfully reaching the grasp (Figure 5a). Another type of successful strategy is to flip the object to stand on its side and then move to the grasp (Figure 5b). This strategy relies on the fact that the object remains stable after the rotation. We will show in the real-robot experiments that for a non-box object, the object may lie on the wall to maintain stability.

## 5.3 Multi-grasp Experiments

**Multi-grasp Training:** Going beyond a single desired grasp, we generate the grasp configurations around the side of the object and parameterized the grasps into a continuous grasp ID in the range of $[0, 4]$ (Figure 6). We train two types of multi-grasp policies with curriculum: *MultiGrasp-Front* which starts from grasp ID=1.5 and *MultiGrasp-Side* which starts from grasp ID=2.5. As a baseline,

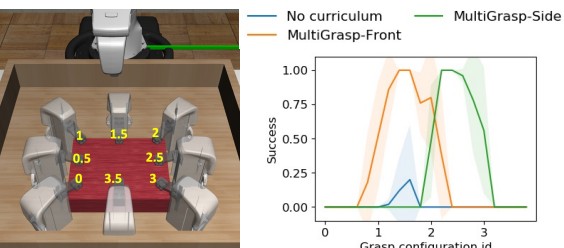

| | MultiGrasp Front | MultiGrasp Side |
|---|---|---|
| ArgmaxQ | $1.00 \pm 0.00$ | $1.00 \pm 0.00$ |
| ArgmaxQ-$t_0$ | $1.00 \pm 0.00$ | $1.00 \pm 0.00$ |
| PoseDiff | $1.00 \pm 0.00$ | $0.96 \pm 0.08$ |
| PoseDiff-$t_0$ | $1.00 \pm 0.00$ | $0.50 \pm 0.43$ |
| Uniform | $0.54 \pm 0.16$ | $0.90 \pm 0.06$ |

Table 1: Comparison of grasp selection methods: Side grasp policies achieve better performance when using the Q-function to select the grasp.

Figure 6: **Left:** Grasp configurations. **Right:** Multi-Grasp Training results with and without curriculum.

we train a policy by uniformly sampling from the entire set of grasps without curriculum (*No curriculum*). Figure 6 shows the performance of these policies evaluated across all grasp IDs. Without curriculum, the agent has difficulties in reaching any of the grasps. With the automatic curriculum, both *MultiGrasp-Front* and *MultiGrasp-Side* expand from a single grasp to most of the grasps on one side of the object. Figures 13a and 13b include qualitative examples of the behaviors which shows that it may require a completely different behavior for different grasps.

**Grasp Selection:** We compare grasp selection methods with *MultiGrasp-Front* and *MultiGrasp-Side*. We sample 50 grasps from the training range of the policy at the beginning of each episode. The grasp selection methods will choose a grasp from this set as the input to the policy. We evaluate the following grasp selection options: *ArgmaxQ* selects the grasp that corresponds to the highest Q-value. *PoseDiff* selects the grasp according to the closest distance to the current gripper pose according to Equation 1b (with the same weights as the reward function). Both *ArgmaxQ* and *ArgmaxQ* select a grasp for each timestep. Alternatively, *ArgmaxQ-$t_0$* and *PoseDiff-$t_0$* only selects a grasp during the first timestep of the episode. *Uniform* samples a grasp from the set uniformly. The results are summarized in Table 1. For *MultiGrasp-Side*, using the Q-function for grasp selection is better than the other approaches. Since the policy has a more complicated maneuver to reach the side (Figure 13b), the Q-function can capture the difficulty of the goal better than pose difference.

### 5.4 Policy Generalization

In this section, we evaluate the generalization of the policy across environment variations: open loop trajectories (*Open Loop*), policies trained over a fixed environment (*Fixed Env*) and policies trained with ADR (*With ADR*). The open loop trajectories are obtained by rolling out the *Fixed Env* policies in the default environment. We sample 100 environments from the range covered by the ADR policies (Appendix C) and plot the percentage of environments that are above a certain performance metric (Figure 7). The closed-loop

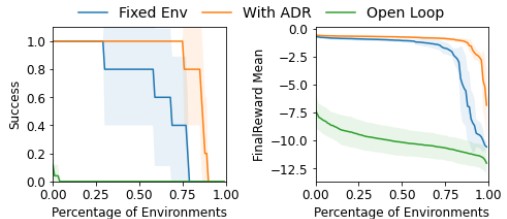

Figure 7: Evaluation on the generalization of the policies by sampling 100 environments.

policies are much better than open-loop trajectories. With ADR, the generalization can be improved even further. Sensitivity analysis on single physical parameters can be found in Appendix A.

### 5.5 Real-robot experiments

We execute the single grasp policies on the real robot with zero-shot sim2real transfer over 10 test cases with different dimensions, densities, surface frictions, and sizes as shown in Figure 8. For non-box objects, the poses are defined with respect to the bounding boxes. The bounding boxes are obtained by running Principle Component Analysis (PCA) on the scanned object point cloud. More details of the real robot experiments can be found in Appendix D. Note that most of the objects are out-of-distribution. We evaluate 10 episodes for each test case and summarize the results in Figure 8. The success is measured by being able to close the gripper and lift the object at the end of the episode. We first compare the policies with and without Automatic Domain Randomization, denoted as **w ADR** and **w/o ADR** respectively. Quantitatively, the policy with ADR achieves a success rate of $78\%$ while the policy without ADR achieves $33\%$. Interestingly, the policy with ADR achieves 24/30 successes over the bottle, the Cool Whip container, and the container with a reversed

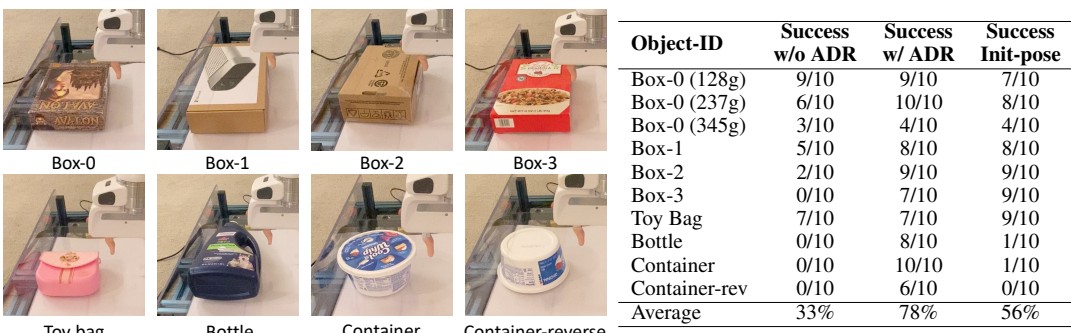

| Object-ID | Success w/o ADR | Success w/ ADR | Success Init-pose |
|---|---|---|---|
| Box-0 (128g) | 9/10 | 9/10 | 7/10 |
| Box-0 (237g) | 6/10 | 10/10 | 8/10 |
| Box-0 (345g) | 3/10 | 4/10 | 4/10 |
| Box-1 | 5/10 | 8/10 | 8/10 |
| Box-2 | 2/10 | 9/10 | 9/10 |
| Box-3 | 0/10 | 7/10 | 9/10 |
| Toy Bag | 7/10 | 7/10 | 9/10 |
| Bottle | 0/10 | 8/10 | 1/10 |
| Container | 0/10 | 10/10 | 1/10 |
| Container-rev | 0/10 | 6/10 | 0/10 |
| Average | 33% | 78% | 56% |

Figure 8: We evaluate the policy on the real robot with various test objects. The policy trained in simulation on box-shape objects can generalize to the real robot and other shapes. With ADR, the policy achieves 45% better success rate.

initial pose. This demonstrates that although the policy is only trained with boxes in simulation, it can also generalize to other shapes to some extent when we represent the object with its bounding box. However, when the object with an out-of-distribution shape has a very different transition dynamics, the policy could fail. Qualitatively, both policies being evaluated exhibit similar strategies as discussed in Section 5.2. In fact, a single policy network may execute either the dropping strategy (Figure 1) or the standing strategy (Figure 5b) depending on the current state. We also include additional results when the initial object location is not close to the wall, denoted as **Init-pose** in Figure 8. We finetune the **w/ ADR** policy to expand further over the range of initial object locations. The success rate remains similar for most objects, but this setting becomes more challenging for non-box shapes. Videos of the full real robot evaluation including failure cases and recovery behaviors can be found on the website [1]. These real robot results are valuable to the field of manipulation because it is beyond what has been shown with a simple hand considering the combined complexity of contact events, object motion and object generalization.

## 6 Limitations

One limitation of this work is that the policy is trained with box-shape objects. Although it may generalize to other shapes to some extent as shown in the experiments, the policy might be improved by including other shapes during training. In addition, the pose of the object alone may not be sufficient to generalize to novel objects; using a better representation of the shape such as a point cloud or key-points could improve generalization across shapes. However, these changes would also increase the training complexity. Another limitation is that we assume a reasonably accurate robot and gripper model, in terms of geometries, kinematic and dynamic parameters. It would be interesting to explore how to extend the method to transfer across robots and grippers.

## 7 Conclusion and Takeaways

In this work, we study the "Occluded Grasping" task where the robot with a parallel gripper aims to reach a grasp configuration using extrinsic dexterity. We present a system that learns a closed-loop policy for this task with reinforcement learning. In the experiments, we demonstrate the importance of each component of the system. We also show that the policies can be executed on the real robot and generalize to various objects. One potential extension of our work is to train the policy with a wide variety of object shapes which may require image-based or point cloud-based policies. Also, the pipeline can potentially be extended to other extrinsic dexterity tasks.

Despite the simplicity of the proposed method, we would like to emphasize the following takeaways from this work: First, we provide a concrete example that a simple gripper can do much more than pick-and-place while being cheaper and easier to maintain than a dexterous hand, following previous work in extrinsic dexterity. We envision more future work in this direction in manipulation. Second, RL can be a good option to generate policies with emergent extrinsic dexterity, and sim2real transfer works reasonably well with our proposed system. Our work takes a step towards deploying contact-rich policies with a simple gripper in the real world.

---

[1] https://sites.google.com/view/grasp-ungraspable

**Acknowledgments**

This material is based upon work supported by the National Science Foundation under Grant No. IIS-1849154 and LG Electronics. We thank Daniel Seita, Thomas Weng, Tao Chen, Homanga Bharadhwaj, and Chris Paxton for the valuable feedback.

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
