# OpenReview forum: "Learning to Grasp the Ungraspable with Emergent Extrinsic Dexterity"
_robot-learning.org/CoRL/2022/Conference — CoRL 2022 Oral_

### Official Review · Reviewer_58SX · 2022-08-01

**Originality:** Excellent
**Technical Quality:** Excellent
**Clarity Of Presentation:** Excellent
**Impact:** 4

**Recommendation:**

Strong Accept: I recommend accepting the paper and will argue for my recommendation even if other reviewers hold a different opinion.

**Summary:**

The paper proposed an RL-based training framework to train a parallel jaw gripper to grasp objects in configurations that necessitate extrinsic dexterity.  Specifically, objects placed on a flat bin are considered. The reward function is designed such that extrinsic manipulations emerges from training without any dedicated reward terms. Detailed experiments demonstrate that a policy trained in simulation with domain randomization can generalize across different objects parameters, and transfers to a physical robot platform without the need for any additional training.

**Issues:**

Please respond to my questions under the "Questions and Suggestions" section above and feel free to push back on any of my comments if they exhibit misunderstanding on my part.

**Quality Of The Limitations Section:**

Limitations are addressed clearly

**Reviewer Expertise:**

4: The reviewer is confident but not absolutely certain that the evaluation is correct

**Robotics Focus:**

Sufficient demonstration on hardware

**Strengths And Weaknesses:**

### Strengths  (in no particular order)

- I thoroughly enjoyed reading this paper. It is expertly written and clearly organized. I consistently found myself asking a question and finding the answer in the very next paragraph.
- The framework leverages a simple but powerful idea: A dense yet simple reward measuring progress when combined with good curriculum design and domain randomization will enable the robot to figure out that the external environment can be used to its benefit.
- Design choices across the framework are justified and appropriately motivated with references to backup claims.
- The trained policy enjoys zero-shot generalization to the real-world, thanks to Automatic Domain Randomization. Further, the objects used in the physical robot experiment were out of distribution.
- The evaluation is thorough and the results conclusively demonstrate the capabilities of the proposed methods and provide insights into the need for different design elements.
- The supplementary website and the videos are very helpful to understand the approach and the learned strategies. I appreciated the discussion of failure modes.

### Questions and suggestions (in no particular order)

- What would happen if the initial location of the object was not in contact with any wall? Will the robot learn to push the object in some direction until contact is made to establish support, or would it simply attempt to lift the object from where it is and fail? I would assume this depends on the training curriculum. But it might be worthwhile to discuss this.
- The paper cites the requirement of a specific gripper design as a limitation of some existing methods. However, the proposed method is trained and tested on the same gripper. How would the proposed approach generalize across grippers or robots? Given that the paper leverage operational space control, it might not be far fetched to investigate if the policy trained on one platform will work on another with some tuning off the controller gains.
- The experiments, while comprehensive and well done as stated above, left me to wonder why the proposed approach was not compared to some of the existing methods (at least the ones that rely on learning such as Refs. [27] and [28] and do not require considerable manual design).
- How sensitive were the choice of weights $\alpha$ and $\beta$ in the reward function, and how much reward engineering was necessary?
- Given that OSC requires a model of the robot, is perfect knowledge of the robot model assumed? This assumption is somewhat buried in the paper and would be better to make it more explicit.
- The grasp selection method involves maximization of the learned Q-function. But it is not clear at what point in time this optimization is carried out. Is it at the beginning before the robot starts its approach?
- Will the source code be made publicly available?

**Summary Of Recommendation:**

The paper presents a simple and elegant learning framework that expands the capabilities of parallel jaw grippers by training policies that implicitly learn to extrinsic manipulation skills. The paper is well writing and the experiments are detailed and provided convincing evidence to support the claims.

---

> ### Author Response · Authors · 2022-08-26
> **Response to Reviewer 58SX (part 1)**
>
> Thank the reviewer for the encouraging comments. We address the questions and suggestions below:
>
> **Q: "What would happen if the initial location of the object was not in contact with any wall? ... I would assume this depends on the training curriculum. But it might be worthwhile to discuss this."**
>
> The reviewer is correct that the initial location of the object depends on the training curriculum. The maximum initial distance between the object and the wall is a parameter in the automatic domain randomization procedure. The policy that we used for the real robot evaluation was only configured to be trained with 0 cm ~ 2 cm “initial distance to wall” as listed in Table 3. However, we performed an additional experiment where we finetuned the previous policy with ADR on the “initial distance to wall” parameter up to 20 cm and then transferred it to the real robot. We updated the videos on the website (https://sites.google.com/view/grasp-ungraspable#h.47gpgvj0b1jv). As can be seen, the policy learns to push the object from the center of the table to the wall.
>
> **Q: "The paper cites the requirement of a specific gripper design as a limitation of some existing methods. However, the proposed method is trained and tested on the same gripper. How would the proposed approach generalize across grippers or robots?"**
>
> Our method is currently designed to be trained and tested on the same gripper. To clarify our point, we use a standard Franka gripper in our experiments, whereas previous work required special grippers to satisfy the specific contact events [2,3] or two grippers with different designs due to the limited action space [11].
>
> As suggested by the reviewer, it would be interesting to explore in the future how to extend the method to transfer across gripper morphologies. This might be achieved by including the geometry of the gripper as an additional input to an image-based or point cloud based policy. We have updated the limitations section to clarify this point.
>
> **Q: Comparison to existing method such as [27] and [28]**
>
> Compared to [27] QT-Opt: This paper proposes an RL algorithm QT-Opt. We use SAC as the underlying RL algorithm but SAC could be replaced by QTOpt, TD3, MPO, etc; the choice of RL algorithm is orthogonal to our contribution of proposing a system for extrinsic dexterity.
>
> Compared to [28] Grasping in the wild: This paper proposes a method for 6DoF closed loop grasping task using human demonstrations with “action-view” based rendering. Their method is not directly applicable to our task due to the following differences: First, our policy is learned from scratch without human data collection, whereas their method relies heavily on demonstration data. Second, we use a continuous action space instead of a discrete action space which affects the choice or RL algorithm; they use Q-learning which is not directly applicable to continuous action spaces. Third, the proposed “action-view” based rendering may not be directly applied to our task. The grasping task considered in this work mainly requires the gripper to reach close to the objects in free space. For such a task, they propose the “action-view” based rendering which renders the virtual next observation of the camera based on the gripper movement action in the free space. However, it is not directly applicable to our task which involves dynamic interaction with the object; rendering a new camera view alone is not sufficient to predict the next state for such dynamic interactions.
>
> **Q: Sensitivity to the choice of weights α and β**
>
> We add more results with different reward weights in Appendix A.3. As can be seen, the policy is not too sensitive to the exact values of the weights.  As a reminder:
> - $\alpha_1$ and $\alpha_2$ weight the translation and rotation error between the target grasp and the current end-effector pos
> - $\beta$ weights the target grasp occlusion penalty which is to penalize the agent if the target grasp configuration is in collision with the table.
>
> The tuning process was very straightforward: when the resulting policy reaches close to the object but doesn’t move the object, this indicates that we need a higher grasp occlusion penalty ($\beta$) to encourage the robot to rotate the object so that the target grasp is not occluded. When the policy only rotates the object but cannot reach the desired grasp, we increase the weights for the pose difference ($\alpha_1$ and $\alpha_2$).

---

> > ### Author Response · Authors · 2022-08-26
> > **Response to Reviewer 58SX (part 2)**
> >
> > **Q: Given that OSC requires a model of the robot, does it assume perfect knowledge of the robot model?**
> >
> > It is true that in principle OSC requires an accurate model of the robot including both kinematic and dynamic parameters. However, we did not explicitly calibrate the robot dynamics in the simulator against our real Franka robot. It is likely that the URDF contains accurate kinematics, but the dynamics (e.g. mass, damping, friction values) are likely not accurate. There is a sim2real gap of the low-level controller due to the inaccurate robot model (as discussed in Appendix D.1 "sim2real gap of the low-level controller") but this gap is partially compensated by using a closed loop policy trained with automatic domain randomizations (action scales in Table 3). We have modified the limitations section to include a discussion on this point.
> >
> > **Q: At what point in time the grasp selection optimization is carried out?**
> >
> > This selection procedure is performed either for each timestep t across the entire episode or at the beginning of an episode. We have updated Section 4.5 to make this more explicit.
> >
> > **Q: Will the source code be made publicly available?**
> >
> > Yes!

---

> > > ### Comment · Reviewer_58SX · 2022-08-26
> > > **Thank you for your thoughtful responses**
> > >
> > > Thank you for considering my comments and incorporating my suggestions. I appreciate the answers and they provide clarity to some previously-fuzzy aspects of the paper. I hope these clarifications are made directly in the revised manuscript. My evaluation of the paper remains positive.

---

### Official Review · Reviewer_Deoh · 2022-08-01

**Originality:** Very Good
**Technical Quality:** Good
**Clarity Of Presentation:** Very Good
**Impact:** 3

**Recommendation:**

Weak Accept: I recommend accepting the paper, but will not argue for my recommendation if the majority of other reviewers have a different opinion.

**Summary:**

This paper proposes a sim-to-real reinforcement learning approach for grasping objects with external dexterity. The specific focus of this paper is to grasp objects lying at the bottom of shallow flat container, from a grasp pose which would cause the gripper body to collide with the container bottom. This is called an occluded grasp. The solution is to push and tip the object against one of the walls of the container, thereby removing the occlusion from the grasp. Experiments are performed in both simulation and on a real robot.

**Issues:**

Not necessary for acceptance, but the paper can be strengthened by including experiments with noisy object tracking.

**Quality Of The Limitations Section:**

Limitations are addressed clearly

**Reviewer Expertise:**

4: The reviewer is confident but not absolutely certain that the evaluation is correct

**Robotics Focus:**

Sufficient demonstration on hardware

**Strengths And Weaknesses:**

## Strengths
- Extrinsic dexterity is under-explored in robot learning. This paper presents a good reference point for environment and reward design and low level robot control.
- The paper is well written and easily understandable.
- Real robot experiments test the validity of simulation results.
- The ablation studies showing the importance of domain randomization and robot control mode (operational space vs. joint space) are useful for readers wanting to draw conclusions about robot RL.

## Weaknesses
- The method depends on live accurate object tracking, because it is a policy observation. However, the paper lacks an analysis of the behaviour of the trained policy with noisy object pose estimates. Noise in different DOFs of the 6-DOF object pose might affect the policy performance differently. Such an analysis would help readers choose object tracking algorithms if they want to replicate this system for their task.

**Summary Of Recommendation:**

I am recommending to accept this paper because of the clear research question and well-executed experiments to answer it. The paper can be further strengthened by addressing the weakness pointed out above.

---

> ### Author Response · Authors · 2022-08-26
> **Response to Reviewer Deoh**
>
> Thank the reviewer for the positive feedback.
>
> **Q: “The paper lacks an analysis of the behaviour of the trained policy with noisy object pose estimates. Noise in different DOFs of the 6-DOF object pose might affect the policy performance differently. Such an analysis would help readers choose object tracking algorithms if they want to replicate this system for their task."**
>
> Thank you for this suggestion; this analysis will indeed be helpful for future readers. We evaluate the policy trained with ADR on different noise levels and include the results in Appendix A.2.
>
> Our object pose estimation method is described in Appendix D.1; to help the reproducibility of our system, we will release the code which includes the implementation of object pose tracker (using the ICP function from open3d). Given a model of the target object and an initial pose estimate, our pose tracker provides a reasonably accurate pose estimation that works for our real robot experiments.

---

### Author Response · Authors · 2022-08-26
**Revised paper and appendix**

**Comment:**

We thank all the reviewers for the helpful feedback. We have revised our paper according to the suggestions and attached it here. All the changes since rebuttal are highlighted in blue. To summarize the major changes during rebuttal:

- Conducted additional real robot experiment when the object starts from the center of the table (https://sites.google.com/view/grasp-ungraspable#h.47gpgvj0b1jv)
- Performed sensitivity analysis on object pose estimation noise (Appendix A.2)
- Added additional training results with different reward term weights (Appendix A.3)
- Included training curves for the ADR experiments (Appendix C)
- Improved clarity on experiment details

**Zip File:**

/attachment/9a1010512d3d37b46ccb1fd217b2296af06d3b06.zip

---

### Meta-Review · Area_Chair_V9nq · 2022-09-04

**Recommendation:** Accept (Oral)
**Confidence:** 5

**Metareview:**

The paper was positively evaluated by 3 reviewers (one as a comment due to emergency reviews). All reviewers aggreed that the proposed task "Occluded Grasping” is new and interesting and the presented method is well motivated. The main issues raised by the reviewers were a better description of the problem formulation as well as the experimental setup and showing the dependency of the algorithm on object tracking errors. Both have been addressed properly.
Due to the positive scores and the innovate problem which is solved in the paper, I recommend an oral presentation.

**Best Paper Nomination:**

No